# Environmental Mismatch: Do Associations between the Built Environment and Physical Activity among Youth Depend on Concordance with Perceptions?

**DOI:** 10.3390/ijerph17041309

**Published:** 2020-02-18

**Authors:** Venurs H. Y. Loh, Jenny Veitch, Jo Salmon, Ester Cerin, Suzanne Mavoa, Karen Villanueva, Anna Timperio

**Affiliations:** 1Institute for Physical Activity and Nutrition (IPAN), School of Exercise and Nutrition Sciences, Deakin University, Geelong 3220, Australia; jenny.veitch@deakin.edu.au (J.V.); jo.salmon@deakin.edu.au (J.S.); anna.timperio@deakin.edu.au (A.T.); 2Mary MacKillop Institute for Health Research, Australian Catholic University, Melbourne 3000, Australia; ester.cerin@acu.edu.au; 3Melbourne School of Population and Global Health, The University of Melbourne, Carlton 3010, Australia; suzanne.mavoa@unimelb.edu.au; 4The Centre for Urban Research, RMIT University, Melbourne 3000, Australia; karen.villanueva@rmit.edu.au

**Keywords:** built environment, MVPA, perception, neighborhood, adolescent

## Abstract

Without accurate awareness of features within the built environment, the availability of a supportive built environment alone may not be sufficient to influence physical activity levels. We examined the moderating effects of concordance/discordance between selected objective and perceived built environment features in the relationship between objective built environment features and physical activity. Cross-sectional data from 465 youth aged 12–20 years from 18 schools in Melbourne, Australia were used. The relationship between trails and physical activity differed by concordance/discordance. There were positive relationships among those with concordant perceptions, but no significant differences for those with discordant perceptions. At least for trails, environmental interventions designed to enhance physical activity may be less effective if efforts are not made to enhance individuals’ awareness of their environment.

## 1. Introduction

Built environments that promote active living are considered important for improving levels of physical activity [1,2]. However, the availability of a supportive built environment alone may not be sufficient to influence PA levels, and individuals’ perceptions of the built environment often vary from the objectively measured environment [3]. Studies in adults have reported that discordance between the perceived and actual built environment (e.g., adults perceiving their neighbourhoods to be less walkable than reality) is common and associated with lower levels of physical activity [4,5,6].

According to the social ecological framework, the built environment-physical activity relationship may depend on individual-level psychosocial factors [7]. However, to date no studies have examined how individuals’ awareness of the built environment influence the relationship between the actual built environment and physical activity. To better understand the relationship between the built environment and physical activity, both objective built environment attributes (environmental level) and concordance between perceptions and actual built environment attributes (individual level) should be considered. 

In particular, adolescents’ concordant (accurate) and discordant (inaccurate) perceptions of the built environment have not been well explored and the way in which perceptions influence associations between the actual built environment and physical activity is not understood. Adolescence is a period when physical activity levels decline steeply when transitioning from childhood [8], and is also a period that is characterised by social changes, including a shift in autonomy away from family to peer interactions [9]. Therefore, the relative importance of built environment features and the mechanisms through which they influence physical activity may not be the same as in adults. 

This study aimed to address this gap by examining whether concordance/discordance between perceived and objective built environment features moderated the relationship between selected objective built environment features and physical activity among adolescents. We hypothesised that the positive effects of objective built environment features on physical activity would be stronger among those with concordance between the perceived and objectively measured built environment. 

## 2. Methods 

Data collected between August 2014 and December 2015 as part of the NEighbourhood Activity in Youth (NEArbY) study conducted among adolescents living in Melbourne, Australia were used for this study. NEArbY is part of the multi-country IPEN Adolescent project (International Physical Activity and the Environment Network Adolescent; http://www.ipenproject.org/IPEN_adolescent.html). Ethical clearance was obtained from relevant education institutions. 

Details about school and participant recruitment across high/low walkability [10] and income areas at statistical area level 1 (SA1) [11] have been published elsewhere [12]. Briefly, 18 of the 137 schools approached agreed to participate. Interested students in year levels nominated by the school received a recruitment package with information about the study, a consent form and parent survey. Written parental consent and student assent were obtained from 528 participants. Of these, 472 wore an accelerometer, 468 students completed an online survey and 287 parents also completed a survey. The residential addresses of 465 students were geocoded. 

## 3. Measures 

### 3.1. Physical Activity 

Physical activity was measured using the ActiGraph GT3X+ (ActiGraph, Pensacola, FL, USA) accelerometer, a valid and reliable instrument for measuring physical activity in youth [13]. Participants were asked to wear the monitor on the hip during waking hours for eight consecutive days. Duration of moderate-to-vigorous-intensity physical activity (MVPA), defined as ≥4 METS/min, was determined using age-appropriate cut-points [14]. On weekdays, MVPA outside of school hours (before school, after school, and evenings [>6 pm]) was computed for days on which participants had ≥50% wear time [15] between the end of school to 6 pm (an indicator of a ‘valid’ day) and averaged for those with at least three valid weekdays. School hours were not considered on weekdays, as it is unlikely that the residential neighbourhood environment influenced physical activity during this time. On weekend days, total duration of MVPA was computed for those with ≥7 h of wear time and averaged across weekend days (min/day).

### 3.2. Age and Sex

Age (years) and sex were self-reported. Missing data on adolescent age were supplemented from responses in the parent survey (*n* = 7). 

### 3.3. Neighbourhood Disadvantage

Neighbourhood disadvantage score was obtained from the Socio-Economic Indexes for Areas based on the Index of Relative Socioeconomic Disadvantage [16] at the SA1 level.

### 3.4. Perceived Built Environment Features

Perceived built environment features were measured using specific items from a modified scale from the Neighbourhood Environment Walkability Scale-Youth (NEWS-Y) [17]. Participants were asked how long it would take them to walk from their home to the nearest park, recreation facility (indoor recreation centre, swimming pool, basketball court, soccer field or skate park) and walking/hiking/biking trails. Response options were (1) 1–5 min, (2) 6–10 min, (3) 11–20 min, (4) 21–30 min, (5) 31 min and above or (5) do not know. Two variables were created for each destination: (1) feature within 1–5 min walk (present vs. absent) and (2) feature within 1–5 or 6–10 min walk (present vs. absent).

### 3.5. Objectively Measured Built Environment 

IPEN Geographic Information Systems (GIS) templates were used to guide the computation of objective indicators of the built environment [18]. Home addresses were geocoded using ESRI ArcGIS 10.3. Street network buffers of 500 m and 1 km were created around each residential address using street centreline data sourced from VicMap Transport [19]. Three variables that were comparable to the perceived built environment features were computed within each of these buffers and each were dichotomised as present or absent:

*Parks:* Parcel count of parks within or intersecting with each buffer were calculated. Parks included protected areas, natural and semi natural areas, parkland and gardens [20,21]. 

*Recreation facilities:* Parcel count of publicly-funded recreational facilities (e.g., soccer fields, basketball courts) compiled from a range of sources [20,21,22]. 

*Trails:* Count of trails within or intersecting with each buffer were identified from State Government of Victoria: Vicmap Transport [19]. 

### 3.6. Concordance/Discordance of Perceived and Objective Built Environment Variables 

An average self-paced casual walk among 13–15 and 16–18 year olds corresponds to 4.2 and 4.4 METS, respectively [23]. Based on the PA compendium for youth [24], the closest match for walking pace at this intensity is three miles per hour (MPH). At a pace of 3MPH, a 5-min walk corresponds to approximately 400 m, and a 10-min walk to 800 m. Therefore, the perceived built environment within a 1–5 min walk (survey) most closely matches the objective built environment measured with a 500 m buffer and the perceived built environment within a 6–10 min walk most closely matches the 1 km buffer. 

For each feature and within each buffer size, ‘’discordance’’ variables were created by comparing perceived and objective data on the presence or absence of a specific feature and coding them as either ‘concordant’ (coded as 0) or ‘discordant’ (coded as 1). Data were coded as ‘concordant’ if the feature was coded as ‘’present’’ in both (e.g., participant reported a park within a 1–5 min walk and ≥1 park was present within the 500 m GIS buffer) or ‘’absent’’ in both (e.g., participant reported no parks within a 1–5 min walk and no parks were present within the 500 m GIS buffer). Data were coded as ‘discordant’ when the perceived absence/presence of the feature did not correspond to the objectively measured absence/presence within the relevant GIS buffer. This included when the feature was perceived to be closer or further than actual. 

## 4. Statistical Analyses

Multilevel logistic regression models were conducted to examine characteristics (age, sex and neighbourhood disadvantage) associated with being concordant/discordant. Separate multilevel linear mixed models were used to examine associations between 500 m and 1 km objectively measured built environment features and MVPA on weekdays and weekend days (min/day). School ID and neighbourhood SA1s were entered as random effect variables to account for cross-classified clustering. Moderating effects of the concordant/discordant variables on associations between the built environment and MVPA were estimated by adding a two-way interaction term to the main effects for each built environment exposure separately. When significant moderation was found, separate effects of the built environment and MVPA were examined for participants with concordant and discordant environmental data. Significant moderation effects are presented graphically (predicted MVPA on weekdays and weekend days plotted against the minimum and maximum of built environment variables at each level of moderator). Data analyses were undertaken using STATA/SE 15.0 (STATA Corp., College Station, TX, USA). 

## 5. Results 

Adolescents’ mean age was 15.3 (SD = 1.5) years, and 59% were girls. On average, participants spent 25.2 min/day (SD = 14.8) in MVPA outside school hours on weekdays and 24.8 min/day (SD = 21.6) in MVPA on weekend days. 

Data on concordance of perceived and objective environmental features are shown in Table 1. In general, a higher percentage of participants had concordant perceptions for built environment features within the 500 m buffer than with the 1km buffer. For example, 76% of participants had concordant perceptions of the presence of trails within 500 m buffer, while 49% of participants had concordant perceptions of trails within the 1 km buffer. Age, sex and neighbourhood disadvantage were not associated with concordance (results not shown). 

On weekdays, no significant associations were found between the selected objective BE features and MVPA (Table 2). Concordance/discordance significantly moderated the relationship between trails within 1 km and weekday MVPA. A significant relationship was observed between trails within 1 km and weekday MVPA among those who had concordant perceptions: for every one-unit increase in trail count within 1 km, MVPA was 5.43 min/day higher (*B* = 5.43; 95% CI 1.83, 9.03; *p* = 0.003). However, no relationship was found among those who had discordant perceptions (*B* = 0.04; 95% CI −2.68, 2.77; *p* = 0.973) (Figure 1a). 

On weekends, objective parks and recreation facilities were significantly positively associated with MVPA (Table 2). Concordance/discordance significantly moderated the relationship between trails within 1 km and weekend MVPA. Among those with concordant perceptions, trails within 1 km were positively associated with weekend day MVPA: for every one-unit increase in trail count within 1 km, MVPA was 6.93 min/day higher (*B* = 6.93; 95% CI 1.63, 12.22; *p* = 0.010). However, no relationship was found among those with discordant perceptions (*B* = −1.13, 95% CI −5.33, 3.06; *p* = 0.597) (Figure 1b). 

## 6. Discussion

This study examined whether concordance/discordance between perceived and objectively measured built environment features moderated associations between objectively measured BE and MVPA among adolescents on weekdays and weekends. No relationships were observed between the selected objectively measured built environment features and MVPA outside school hours on weekdays, but positive relationships between the number of parks and recreation facilities and MVPA on weekends were present. The relationship between trails (1 km) and MVPA on weekdays and weekend days differed by concordant/discordant perceptions: positive associations were found among those with concordant perceptions but no associations were observed among those with discordant perceptions. 

Consistent with previous studies [4,5,25,26,27], discordance between perceived and objectively measured built environment measures was common, particularly for recreation facilities. Adolescents may be unaware of recreation facilities that they do not use. However, discordance may also reflect difficulties in accurately estimating time-to-walk to destinations or variation in time-to-walk within samples. For example, individuals may walk at a faster or slower pace than estimated, so calculated time-to-walk may not align accurately with buffers used. Further, the constructs of perceived and objectively measured built environment may be different (i.e., perceived built environment survey questions may not adequately reflect the objective built environment measures). 

We found significant main effects of parks and recreation facilities on weekend MVPA, but no main effect was found between built environment features and weekday MVPA. This finding suggests that less structured weekend days may encourage MVPA around home. Even though the effect sizes were small, investing in supportive built environment features is likely to have a long-term positive effect on physical activity given the potential to reach large numbers of the population. We also found that the concordance of trails moderated the relationship between trails within 1km and MVPA. Trails have the potential to provide a broader range of opportunities to accumulate MVPA through both active transportation and leisure activities than parks and recreation facilities [28]. 

The lack of focus on the quality of destinations and/or limiting exposure to the residential neighbourhood may help explain some of the null findings in this study [29]. Adolescents may travel greater distances (>1 km) to visit parks or recreation facilities for MVPA than visiting the nearest facility, which may be of a poorer quality or have features that do not meet their needs [30,31]. Alternatively, those who rarely walk to or use neighbourhood destinations may have low awareness of their existence or location. Further, the BE around home may not be the single most important context for adolescents’ PA, as adolescents also engage in PA around school or during the journey to school from home [32]. 

The cross-sectional design means that claims about causality cannot be made. Only a crude measure of concordance was used rather than measuring the degree of concordance. The self-report measures were not purposively designed to address concordance of perceptions and, therefore, the perceived time-to-walk to the selected destinations may not be an optimal match to the buffers used. With the exception of parks, those who had discordant views typically perceived trails and recreation facilities to be closer than actual, but we were unable to test interactions at this level due to low frequencies in responses. Finally, the MVPA measure lacked specificity; activities undertaken within the residential neighbourhood and elsewhere could not be distinguished. 

## 7. Implications

Our findings imply that multilevel strategies that targeting both awareness of trails among individual (through signage or public mass media program) and provision of supportive trail infrastructure for physical activity may be important for encouraging physical activity among adolescents. 

## 8. Conclusions

In conclusion, this study highlights that, at least for trails, environmental interventions designed to enhance MVPA may be less effective if efforts are not made to enhance individuals’ awareness of their environment.

## Figures and Tables

**Figure 1 ijerph-17-01309-f001:**
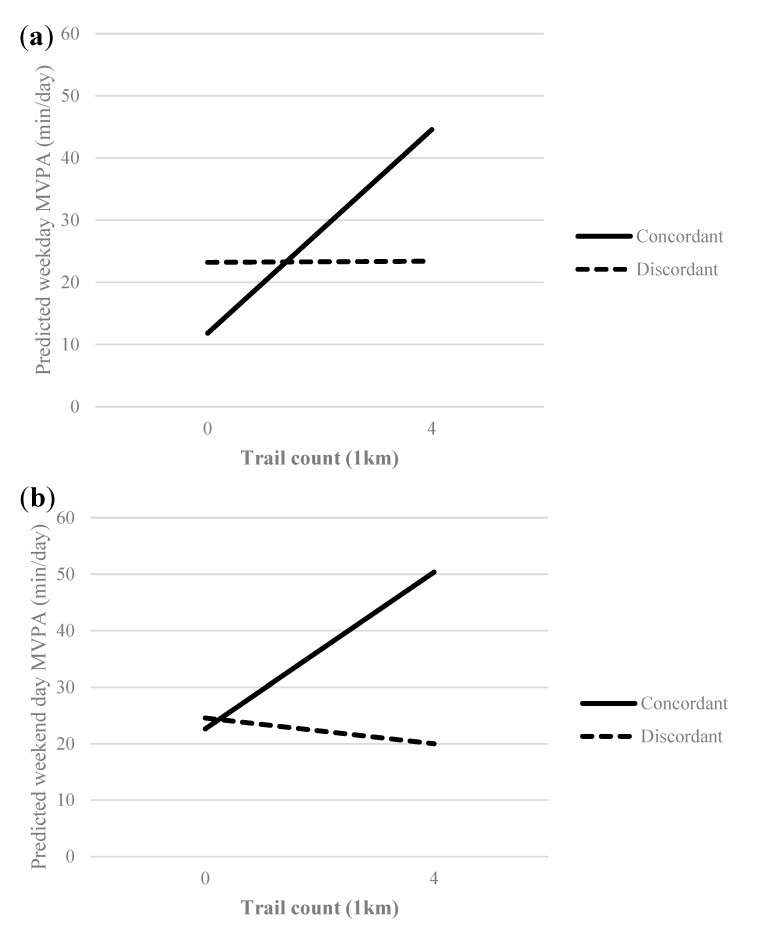
Marginal means plot of significant interactions between trails within 1 km and moderate-to-vigorous physical activity (MVPA) on (**a**) weekdays and (**b**) weekend days according to concordant/discordant perceptions.

**Table 1 ijerph-17-01309-t001:** Built environment variables and the proportion of participants with concordant and discordant perceptions.

	Built Environment					
	**Park (*n* = 463)**		**Recreation Facilities (*n* = 463)**		**Trails (*n* = 463)**	
	**500 m**	**1 km**	**500 m**	**1 km**	**500 m**	**1 km**
**Objective built environment, *n* (%)**						
Present	418 (90.3)	456 (98.5)	138 (29.8)	294 (63.5)	67 (14.5)	107 (30.2)
Absent	45 (9.7)	7 (1.5)	325 (70.2)	169 (36.5)	396 (85.5)	247 (69.8)
	**Park (*n* = 463)**		**Recreation Facilities (*n* = 464)**		**Trails (*n* = 460)**	
**Perceived built environment, *n* (%)**	**1–5 min**	**1–5/6–10 min**	**1–5 min**	**1–5/6–10 min**	**1–5 min**	**1–5/6–10 min**
Present	291 (62.8)	398 (85.9)	329 (70.9)	423 (91.2)	98 (21.3)	228 (49.6)
Absent	172 (37.2)	65 (14.1)	135 (29.1)	41 (8.8)	362 (78.7)	232 (50.4)
**Concordant/discordant (two categories), *n* (%)**	**Park (*n* = 461)**		**Recreation Facilities (*n* = 462)**		**Trails (*n* = 458)**	
Concordant	297 (64.4)	288 (62.5)	197 (42.6)	149 (32.2)	346 (75.5)	223 (48.7)
Discordant	164 (35.6)	173 (37.5)	265 (57.4)	313 (67.8)	112 (24.5)	235 (51.3)
**Concordant/discordant (four categories), *n* (%)**						
Concordant						
Present/present	271 (58.8)	287 (62.3)	100 (21.6)	134 (29.0)	26 (5.7)	50 (10.9)
Absent/absent	26 (5.6)	1 (0.2)	97 (21.0)	15 (3.2)	320 (69.9)	173 (37.8)
Discordant						
Perceived closer than actual	19 (4.1)	19 (4.1)	228 (49.3)	265 (57.4)	71 (15.5)	160 (34.9)
Perceived further than actual	145 (31.5)	154 (33.4)	37 (8.1)	48 (10.4)	41 (8.9)	75 (16.4)

**Table 2 ijerph-17-01309-t002:** Associations between the objective built environment features and moderate-to-vigorous physical activity (MVPA) and the moderating effects of concordance/discordance between perceived and objectively measured presence or absence of built environment features (*n* = 354).

Built Environment Feature	Weekday MVPA (min/day) *n* = 412		Weekend MVPA (min/day) *n* = 356	
	500 m B (95% CI)	1 km B (95% CI)	500 m B (95% CI)	1 km B (95% CI)
**Park (*n*) ^1^**	−0.10 (−0.51, 0.30)	0.02 (−0.12, 0.17)	0.23 (−0.34, 0.81)	0.22 (0.01, 0.43) *
**Interaction ^2^:**				
**x discordant ^3^**	−0.09 (−0.97, 0.79)	0.25 (−0.07, 0.58)	0.27 (−1.07, 1.61)	0.27 (−0.19, 0.75)
**Recreation facility (*n*) ^1^**	−0.53 (−2.64, 1.56)	0.41 (−0.46, 1.28)	2.29 (−0.73, 5.32)	1.69 (0.47, 2.91) **
**Interaction ^2^:**				
**x discordant ^3^**	1.52 (−2.92, 5.97)	1.67 (−0.03, 3.37)	4.86 (−1.71, 11.45)	1.11 (−1.31, 3.54)
**Trails (*n*) ^1^**	0.57 (−2.93, 4.08)	1.78 (−0.46, 4.02)	-0.01 (−5.17, 5.17)	1.83 (−1.50, 5.18)
**Interaction ^2^:**				
**x discordant ^3^**	−4.53 (−11.80, 2.73)	−5.38 (−9.76, −1.01) *	−3.33 (−14.38, 7.71)	−8.06 (−14.72, −1.40) *

All models adjusted for age, sex and neighbourhood disadvantage; ^1^ Main effect model; ^2^ Model including main effects and interaction term between built environmental feature and concordance/discordance of environmental measures; ^3^ Reference group = concordant. * *p* < 0.05; ** *p* < 0.01

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
