# Peer review of "Environmental Mismatch: Do Associations between the Built Environment and Physical Activity among Youth Depend on Concordance with Perceptions?"

_ijerph, 2020, doi:10.3390/ijerph17041309_

Round 1
Reviewer 1 Report
Results
Readers will need more explanation regarding Table 1. To do so, please add a couple of “for examples” after the sentence, “In general, a higher percentage of participants with concordant data were observed for built environment features within 500m than the 1km buffer.”
Page 4, paragraphs one and two: Please explain for readers the meaning of the beta coefficient(s) with respect to more or less MVPA for the concordant coded response and also explain the meaning of the beta coefficient(s) with respect to more or less MVPA for the discordant coded response:
What does a positive beta mean with respect to accelerometer data for the concordant response? What does a negative beta mean with respect to accelerometer data for the concordant response? What does a positive beta mean with respect to accelerometer data for the discordant response. What does a negative beta mean with respect to accelerometer data for the discordant response?Author Response
Please see attachment

Reviewer 2 Report
The authors present an interesting analysis of perceived versus measured built environments and physical activity among Australian youth. They found few associations with weekday MVPA, but several associations for weekend MVPA. Only trails showed significant moderation by perceived presence. This is an interesting paper and contributes to the limited literature on perceptions versus reality of built environment features and physical activity.
Concerns:
1. What was the distribution of MVPA minutes? These are often highly skewed and could benefit from log transformation before regression.
2. Related to #1 above, I assume, as written, that MVPA was not transformed. Therefore, I understand the Beta coefficients in Table 2 should be interpreted as differences in minutes per day of MVPA. If that is correct, the very small effect sizes may warrant some mention in the discussion. (e.g. 0.22 min/day for parks, 1.69 min/day for rec centers)
3. The first paragraph under "Implications" may overstep the results of this study. This study only presents evidence that awareness campaigns might be useful for trails.
Suggestion:
Table 2 could make a compelling figure if predicted minutes/day were generated in place of beta coefficients. The main effect could be plotted along side the concordant/discordant estimates. minutes/day would be more relatable than beta coefficients.
